# The importance of effect sizes when comparing cycle threshold values of SARS-CoV-2 variants

Celine Brinkmann[1,2], Peter Gohl[3], Dietrich Mack[3], Johannes Pfeifer[4], Mònica Palmada Fenés[2], Oliver Harzer[3,5], Bernhard Zöllner[1] *

1 Bioscientia Laboratory MVZ Nordrhein, Moers, Germany, 2 Rhine-Waal University of Applied Sciences, Kleve, Germany, 3 Bioscientia Laboratory MVZ Ingelheim, Ingelheim, Germany, 4 Rhine-Waal University of Applied Sciences, Kamp-Lintfort, Germany, 5 Danube Private University (DPU), Krems an der Donau, Austria

* bernhard.zoellner@bioscientia.de

**Data Availability Statement:** All relevant data are within the paper and its Supporting Information files.

## Abstract

### Purpose

We aimed to elaborate whether cycle threshold (Ct) values differ significantly between wild type SARS-CoV-2 (wtV) and certain viral variants and how strong or weak a potential significant effect might be.

### Methods

In a retrospective study, we investigated 1873 SARS-CoV-2 positive samples for the occurrence of viral marker mutations. Age, gender, clinical setting, days after onset of symptoms, and Ct values were recorded. Statistical analysis was carried out with special consideration of effect sizes.

### Results

During the study period wtV was detected in 1013 samples (54%), while 845 (45%) patients carried the Alpha variant of concern (VOC), and 15 (1%) the Beta VOC. For further analysis, only wtV and the Alpha VOC were included. In a multi-factor ANOVA and post-hoc test with Bonferroni-correction for the age groups we found significant main-effects for Ct values of the viral variant (wtV mean 26.4 (SD 4.27); Alpha VOC mean 25.0 (SD 3.84); $F_{(1,1850)} = 55.841$; $p < .001$) and the clinical setting (outpatients: mean 25.7 (SD 4.1); inpatients: mean 27.0 (SD 4.2); $F_{(1,1850)} = 8.520$, $p = .004$). However, since the effect sizes were very small (eta squared for the Alpha VOC = .029 and the clinical setting = .004), there was only a slight trend towards higher viral loads of the Alpha VOC compared to wtV.

### Conclusions

In order to compare different variants of SARS-CoV-2 the calculation of effect sizes seems to be necessary. A combination of p-values as estimates of the existance of an effect and

**Funding:** The author(s) received no specific funding for this work.

**Competing interests:** The authors have declared that no competing interests exist.

effect sizes as estimates of the magnitude of a potential effect may allow a better insight into transmission mechanisms of SARS-CoV-2.

## Introduction

The new pandemic coronavirus SARS-CoV-2 has the capacity to select for mutational variants. Particularly variants of concern (VOC) pose a considerable threat because they exhibit increased transmissibility, detrimental change in COVID-19 epidemiology, an increased virulence or change in clinical disease presentation or a decreased effectiveness of public health and social measures or available diagnostics, vaccines and therapeutics [1, 2].

Currently, five VOCs have been described in the European Union: the Alpha, Beta, Gamma, Delta, and Omicron variant of concern [3]. In January 2021, almost exclusively wild-type SARS-CoV-2 (wtV) circulated in Germany. However, in the further course, it was more and more replaced by the Alpha VOC, reaching a 88% prevalence of this variant at the end of March 2021 [4]. During this period, it was possible to compare the properties and kinetics of wild type and mutated SARS-CoV-2 in a direct way.

Cycle threshold (Ct) values that are generated in a real-time polymerase chain reaction (PCR), are a way of estimating the virus concentration in a sample semi-quantitatively. This roughly reflects the viral load in the patient's material and could allow conclusions to be drawn about the infectiousness of a patient [5].

The Ct values of the individual VOCs were examined and compared in some studies [6–9]. The results of these investigations were that significant differences between Ct values of VOCs could be determined. However, statistical significance is only one part of the equation. While statistical significance shows that an effect exists in a study and is unlikely to be caused by chance, effect sizes describe whether the effect is large enough to be meaningful in the real world [10]. Particularly with large sample sizes it is more likely that significant results with low p-values can be found in quantitative studies [11]. Therefore, reporting both p-values and effect sizes is much more informative and relevant for the situation *in vivo* than reporting only p-values. For this purpose, effect sizes can be calculated within different statistical models. In ANOVA, eta squared ($\eta^2$) is well suited for the calculation of the effect size and gives an estimate how strong the variation, e.g. of the viral load, can be accounted to each factor or predictor. This makes it possible to classify which factor is more or less important.

Our study aimed to investigate whether there was a significant difference in Ct values between wild-type SARS-CoV-2 (wtV) and its mutational variants of concern. Particularly the influence of multiple cofactors like age, gender, days from onset of symptoms, and clinical setting on Ct values were analysed with special consideration of the magnitude of the effect.

## Material and methods

### Subjects and samples

The study period of this retrospective analysis lasted from January 20th until March 17th 2021. All positive samples in the even-numbered weeks, regardless of the measured Ct-values, and a collective of samples with a Ct value <30 in the uneven numbered weeks were included. Dry swabs without transport medium were taken from the patients´ naso- or oropharyngeal area, shipped to our laboratory within 24–48 hours at room temperature and processed within a mean time of 22 hours. For all patients, age, gender, time of sampling, in- or outpatient status,

and Ct values were documented. In telephone calls with the institutions that sent positive samples, the time between patient´s onset of symptoms and sampling was determined.

### SARS-CoV-2 PCR and determination of SARS-CoV-2 VOCs

PCR was run on a cobas® 8800 instrument (Roche Diagnostics, Mannheim, Germany) using the cobas® SARS-CoV-2 test (also Roche Diagnostics, Mannheim, Germany), which includes specific detection of the *ORF1 a/b* non-structural region and *E*-gene. Pool tests were not performed, each sample was tested individually. To ensure (i) the stability of the viral RNA during transport and (ii) the reproducibility of our PCR system we investigated three previously tested samples with different Ct values. These samples were stored at room temperature and retested on three consecutive days. Thereafter, the day-to-day coefficients of variation were calculated for each of these samples.

SARS-CoV-2 mutational variants were analysed with the cobas® SARS-CoV-2 Variant Set-1, which enabels the detection of the key mutations for the Alpha-, Beta-, and Gamma-VOCs (N501Y, E484K, and the 69–70 deletion).

### Statistical methods

The Mann-Whitney U-test was applied in a first attempt using Abacus 2.0 (Labanalytics, 2021, Jena, Germany) to analyse Ct values in females and males as well as in- and outpatients and different age groups, as described previously [e.g. references 5 and 6]. Using this test, the Ct values of wtV and the Alpha VOC for the respective groups were compared. Thus, two groups were formed for these demographic factors, each consisting of unpaired samples. Similarly, rates of hospitalized patients with wtV and the Alpha VOC were compared with Fisher´s exact test.

To avoid alpha-error-accumulation which occurs with e.g. multiple testing we ran a multi-factor Analysis of Variance (ANOVA)(IBM SPSS for Windows, Version 27.0, 2020, Armonk, NY, USA) afterwards and compared the results of both analyses. The ANOVA model term contained exclusively the main effects, no interactions were included. The factor "days after onset of symptoms" was excluded due to a low sample size. All the assumptions for the calculation, including equality of variances ($F = 1.026$; $p = .427$) were met. Post-hoc-test with Bonferroni-correction was calculated for the age factor, as it had more than two groups. A p-level of $<0.05$ was considered to be statistically significant.

The effect sizes were calculated with the designated standard function of SPSS (see in S1 Table). Values of $\eta^2$ between 0.01 and 0.059 are considered as small effects, those between 0.06 and 0.139 correspond to a medium effect, and of 0.14 or greater to a large effect.

### Results

A total of 1873 SARS-CoV-2 positive samples were analysed with Ct values ranging between 13–36. The reproducibility and stability of Ct values over a period of three days at room temperature was high: repeated measurements of three samples on three consecutive days showed a coefficient of variation of 3% (sample 1, Ct value 17), of 2.2% (sample 2, Ct value 23), and of 2.3% (sample 3, Ct value 28).

Wild type virus was detected in 1013 samples (54%), 845 (45%) patients carried the Alpha VOC, and 15 (1%) the Beta VOC. The distribution of patients between outpatients and inpatients was 954/59 for wtV, 817/28 for the Alpha VOC, and 15/0 for the Beta VOC, respectively. The prevalence of the Alpha VOC changed from 9% (41/464) in calendar week 4 to 74% (406/552) in calendar week 10 (Fig 1). The Beta VOC played a subordinate role with 0.9% (4/438) in week 8 and 2% in week 10 (11/552) (Fig 1).

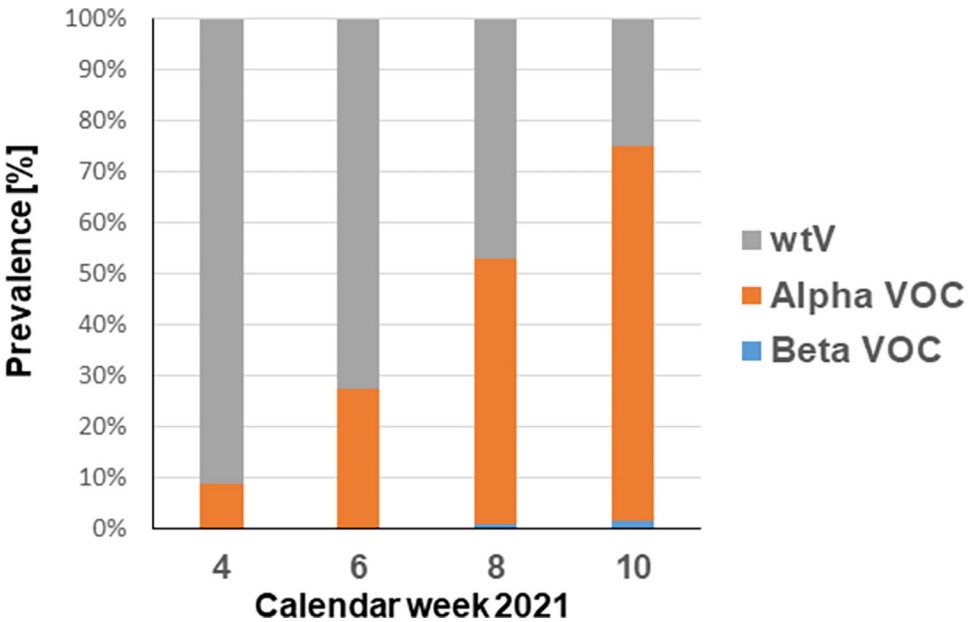

**Fig 1. Prevalence of all positives of wild type SARS-CoV-2 (wtV) and the Alpha and Beta VOCs.**

Due to the low case number (n = 15), the Beta VOC was excluded from further analysis. Table 1 shows the results of the remaining samples (n = 1858) using a Mann-Whitney-U two-group comparison as described in other publications [e.g. references 5 and 6]. Significantly, lower Ct values of the Alpha VOC were detected for the factor gender (p<0.0001), outpatients (p<0.0001), and age groups 0–80 years (p<0.004). The rate of inpatients carrying the Alpha VOC (28/845) was significantly lower than those carrying wtV (59/1013, p = 0.011). The time between the onset of symptoms and the day of sampling could be determined cross-sectionally

**Table 1. Ct values and standard deviations (SD) of wild type SARS-CoV-2 (wtV) and the Alpha variant of concern (VOC) with p-values derived from the associated Mann-Whitney-U tests.**

| Factor (cases: wtV/Alpha VOC) | | Mean Ct value (SD) | | p-value |
|---|---|---|---|---|
| | | **wtV** | **Alpha VOC** | |
| **All** | (1013/845) | 26.4 (4.27) | 25.0 (3.84) | <0.0001 |
| **Gender** | Female (534/406) | 26.4 (4.32) | 25.2 (3.86) | <0.0001 |
| | Male (479/439) | 26.5 (4.21) | 24.9 (3.82) | <0.0001 |
| **Setting** | outpatient (954/817) | 26.4 (4.25) | 24.9 (3.83) | <0.0001 |
| | inpatient (59/28) | 26.9 (4.57) | 27.2 (3.47) | 0.73 |
| **Age [years]** | 0–20 (116/123) | 26.9 (4.44) | 25.3 (3.93) | 0.003 |
| | 21–40 (296/290) | 26.6 (4.19) | 25.2 (3.67) | <0.0001 |
| | 41–60 (361/291) | 26.2 (4.19) | 24.5 (3.82) | <0.0001 |
| | 61–80 (176/120) | 26.4 (4.32) | 24.9 (4.04) | 0.004 |
| | >80 (64/21) | 25.3 (4.46) | 25.6 (4.33) | 0.8 |
| **Days from onset of symptoms (subgroup)** | 1 (26/63) | 26.8 (3.54) | 25.0 (3.47) | 0.03 |
| | 2 (19/51) | 24.4 (3.32) | 24.4 (3.72) | 0.96 |
| | 3 (19/22) | 23.7 (3.11) | 25.0 (4.11) | 0.24 |
| | 4 (9/19) | 26.4 (3.60) | 24.4 (3.32) | 0.16 |
| | 5 (5/9) | 24.2 (4.15) | 24.2 (3.15) | 0.99 |

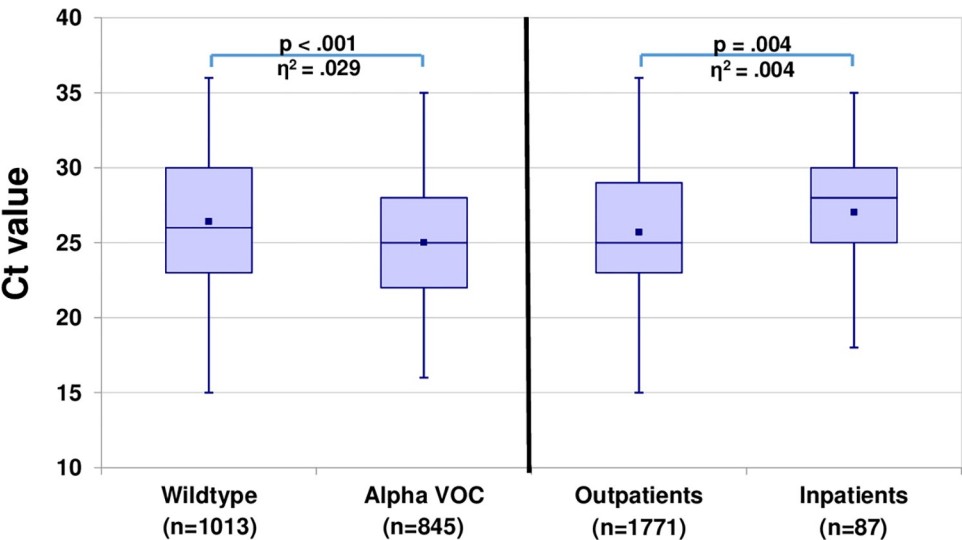

**Fig 2. Box-Whisker-Plots of Ct values for SARS-CoV-2.** (Left) Type of SARS-CoV-2. (Right) Clinical setting. Eta squared ($\eta^2$) indicates the effect size.

for a subgroup of 242 patients until day 5 of the infection. The results revealed only a weak significance on day one showing slightly lower Ct values of the Alpha VOC compared to wtV (25.0 (SD 3.47) *vs.* 26.8 (SD 3.54), p = 0.03) (Table 1). Days 2–5 showed no significant differences in Ct values between wtV and the Alpha VOC. Hence, this parameter was excluded from the subsequent multiple factor analysis.

To avoid the multiple comparison problem of testing, and further investigate the relevance of all factors together, we employed a multi-factor ANOVA. This model showed significant main effects on Ct values for the factors viral variant ($F_{(1,1850)}$ = 55.841, p < .001), age ($F_{(4,1850)}$ = 3.297, p = .011), and setting ($F_{(1,1850)}$ = 8.520, p = .004). The gender groups did not differ significantly regarding the viral load. The effect sizes showed a small effect for the Alpha VOC ($\eta^2$ = .029) as well as for the setting ($\eta^2$ = .004) and age ($\eta^2$ = .007), meaning that e.g. 2.9% of the variation in the Ct values account for the factor viral variant. Wildtype virus had slightly higher Ct values (mean 26.4, SD 4.27) compared to the Alpha VOC (mean 25.0, SD 3.84) and total inpatients had slightly higher Ct values (mean 27.0, SD 4.2) than total outpatients (mean 25.7, SD 4.12) (Fig 2). Post-hoc tests with Bonferroni-correction showed no significant differences between the age categories, so the small, significant main effect was neglected. Taken together, significant differences of Ct values could be found between wtV and the Alpha VOC and between each total in- and outpatients, however with only very small effect sizes.

## Discussion

The investigation of viral variants of SARS-CoV-2 is becoming increasingly important. There has been a great effort to characterize and compare wild type SARS-CoV-2 and its mutational variants *in vitro* [12–17]. In these studies, increased infectivity and faster viral entry into target cells have been shown for VOCs compared to wtV. However, studies on the comparison *in vivo* are difficult to perform as wtV and its variant should circulate in the population at the same time. That is why there are only a few analyses with this approach [6–9, 18–22].

In our analysis, patients carrying the Alpha VOC had a mean Ct value that was one PCR cycle lower than patients carrying wtV, which is in line with previously published studies [6,

7]. Also, in the study by Luo et al. [8] the mean difference in Ct values between the B.1.2, Delta and Alpha variants was in the range of 1–2 PCR cycles (20.61 vs. 19.62 vs. 21.77, respectively). Although these and our results were statistically highly significant, the mean difference of 1–2 PCR cycles is low. P-values, however, are confounded by large sample sizes [11]. This could have been true for at least our study and was the reason why we calculated effect sizes. Also, there is another problem, namely the lack of standardization of the pre-analytical phase prior to the PCR for SARS-CoV-2. For instance, the technique of swabbing or the influence of transport conditions on individual samples could cause a bias in the evaluation of Ct values. One possible solution to balance out preanalytical errors in single samples is to analyse large patient groups, as it was done in our study. However, as mentioned above, this requires the specification of effect sizes in addition to the analysis of p-values.

We found lower Ct values of the Alpha VOC in both sexes, in outpatients and almost all age groups. However, a subsequent multiple-factor analysis led to the result that there was only a significant difference between wtV and the Alpha VOC in the total cohort and between in- and outpatients concerning Ct values and, of great importance, that the effect size was low for these factors. Particularly, the low influence of the viral variant on Ct values was surprising, considering the observation of higher transmissibility of the Alpha VOC compared to wtV [6–9, 18–22]. A plausible explanation could be an increased binding affinity of this variant to the ACE2 receptor [14–16]. This leads to higher infectivity even with comparable virus concentrations [17].

Our study covered a period during which the Alpha VOC had almost completely replaced the wtV. The prevalence rates in our cohort correlate well to the general German prevalence rates determined by the Robert Koch Institute [4]. In addition, hardly any people were vaccinated at that time in Germany. Therefore, a direct comparison of Ct values in currently taken samples was possible and the results were not biased by different immune protection rates of infected persons.

Because the time since symptom onset is a key determinant of viral load in SARS-CoV-2 infections [23, 24], we tried to control for this cofactor. A limitation was, that we could only retrospectively record the days between onset of symptoms and sampling in a cross-sectional manner and that no longitudinal examination of the Ct values could be carried out in individuals during the acute phase of the infection. Nevertheless, our results implicate that apart from slightly lower Ct values of the Alpha VOC at day one there were no significant differences between wtV and VOC in the Ct values even during the first five days of the disease. This strengthens the hypothesis that the increased infectiousness of the Alpha VOC *in vivo* relies on other factors rather than high viral concentrations during the initial phase of the infection process. *In vivo*, factors like receptor-affinity [14–16] or enhanced target cell entry [12, 15] e.g. associated with altered spike cleavage sites [12] of the Alpha VOC seem to play a more important role.

## Conclusions

For future studies elaborating Ct values of e.g. the Delta, Omicron (in particular BA.2 or BA.5) or other emerging VOCs, statistical analyzes with multiple factors seem to be essential. Additionally, we propose the calculation of effect sizes, which would allow an estimation of the magnitude of differences in Ct values and of their true meaning *in vivo*. This could provide clues to the really decisive factors in such a complicated, multifactorial process as the infectivity and transmission of SARS-CoV-2.

## Supporting information

**S1 Table. Output of effect sizes in SPSS for the data of our study.**
(PNG)

**S1 Dataset. Anonymized original dataset.**
(PDF)

## Author Contributions

**Conceptualization:** Celine Brinkmann, Bernhard Zöllner.

**Data curation:** Celine Brinkmann, Johannes Pfeifer, Bernhard Zöllner.

**Investigation:** Celine Brinkmann, Peter Gohl, Dietrich Mack, Johannes Pfeifer, Oliver Harzer, Bernhard Zöllner.

**Methodology:** Celine Brinkmann, Peter Gohl, Dietrich Mack, Oliver Harzer, Bernhard Zöllner.

**Supervision:** Mònica Palmada Fenés, Bernhard Zöllner.

**Writing – original draft:** Celine Brinkmann, Johannes Pfeifer, Bernhard Zöllner.

**Writing – review & editing:** Mònica Palmada Fenés.

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
