## [Decision Letter · Decision Letter 0]

7 Jun 2022

PONE-D-22-07729The importance of effect sizes when comparing cycle threshold values of SARS-CoV-2 variantsPLOS ONE

Dear Dr. Zöllner,

Thank you for submitting your manuscript to PLOS ONE. After careful consideration, we feel that it has merit but does not fully meet PLOS ONE’s publication criteria as it currently stands. Therefore, we invite you to submit a revised version of the manuscript that addresses the points raised during the review process. Unfortunately, one of the two independent reviewers did not provide his comments.

We look forward to receiving your revised manuscript.

Kind regards,

Etsuro Ito

Academic Editor

PLOS ONE

Journal Requirements:

Reviewers' comments:

Reviewer's Responses to Questions

**Comments to the Author**

1. Is the manuscript technically sound, and do the data support the conclusions?

Reviewer #1: Yes

2. Has the statistical analysis been performed appropriately and rigorously? 

Reviewer #1: Yes

3. Have the authors made all data underlying the findings in their manuscript fully available?

Reviewer #1: No

4. Is the manuscript presented in an intelligible fashion and written in standard English?

Reviewer #1: Yes

5. Review Comments to the Author

Reviewer #1: The study compares the Ct, a direct indicator of the viral load, of the wild type SARS-CoV-2 and and its mutational variants. In particular, it insists on the need to carry out the analysis of effect size in order to rigorously assess the possible differences.

The effect size which is the central statistic and the fundamental element of this publication is not explained enough. The introduction should give a sufficient ruling on this question and above all define what the size effect means and its important contribution to the resolution of scientific questions.

The statistics used should be better defined and explained. It is also important to indicate the software used, the versions and the years to carry out the statistical tests. For this section, also specify the level of significance for all the statistical tests carried out.

1) Line 101 How long were samples stored and in what media. Please precise the storage conditions because it can affect the stability of the virus genetic material and consequently the ct-values.

2) Line 109 Please were samples tested in pools or singly? Why did authors not include oropharyngeal samples?

3) Line 104. Can't transporting samples at room temperature alter the quality of the genetic material and make the results uncertain in favor of the samples that would have survived the temperature better?

4) Line 110, indicate manufacturer and country for cobas® 8800 instrument

5) Line 112: Write ORF1ab and E gene in italics

6) Line 129-129, it is not only the age factor that has more than 2 groups, the setting factor also

7) Line 123. Please clarify if the comparison of patient percentages between Wild, Alpha and Beta types SARS-CoV-2 is for hospitalized patients only? Does this include only in-patients or also out-patients?

8) Line 134. Why still talk about Ct range between 15-36 when it had been declared that only samples with Ct value <30 are included in the study (line 102). Please clarify

9) Line 139. The addition of all the cases (1013 wild type, 845 alpha VOC, 15 beta VOC) gives 1901 positive samples and not 1873 as declared in line 134. Please clarify

6. PLOS authors have the option to publish the peer review history of their article (what does this mean?). If published, this will include your full peer review and any attached files.

Reviewer #1: No

---

## [Author Response · Author response to Decision Letter 0]

6 Jul 2022

Dear Prof. Ito,

thank you for giving us the opportunity to resubmit a revised version of our manuscript (manuscript no. PONE-D-22-07729) entitled 

“The importance of effect sizes when comparing cycle threshold values of SARS-CoV-2 variants” by Celine Brinkmann, Peter Gohl, Dietrich Mack, Johannes Pfeifer, Mònica Palmada Fenés, Oliver Harzer and Bernhard Zöllner*

We would also like to thank the reviewer for the valuable comments. We very much appreciate the suggestions for improvement of our manuscript and revised the article according to the comments: 

Have the authors made all data underlying the findings in their manuscript fully available?

Reviewer #1: No

We uploaded all anonymized data underlying the study when submitting the first version of the article (see file “Article_Brinkmann et al._data set.pdf”). Did something get lost?

Major point A:

„The effect size which is the central statistic and the fundamental element of this publication is not explained enough. The introduction should give a sufficient ruling on this question and above all define what the size effect means and its important contribution to the resolution of scientific questions.“

We now have added a paragraph into the introduction in which the problems of p-values and the advantage of reporting effect sizes are described (lines 88-101). In addition, the type of calculation and interpretation of the effect size is now described in the Material and Methods section (lines 152-155 and supporting information Table S1).

Major point B:

„The statistics used should be better defined and explained. It is also important to indicate the software used, the versions and the years to carry out the statistical tests. For this section, also specify the level of significance for all the statistical tests carried out.“

The statistics were defined and described in more detail in the section Material and Methods including software versions and years of release (lines 135-145). Also, the level of significance has been specified (lines 150-151) and the cut off values for the interpretation of eta squared have been inserted (lines 153-155). We have also included the printout of the ANOVA with the respective effect sizes as a supporting information (S1 Table, see also point 9) at the end of the manuscript. This makes our approach clearer for the reader.

Point 1) „How long were samples stored and in what media. Please precise the storage conditions because it can affect the stability of the virus genetic material and consequently the ct-values.“

This is an important point. Swabs without transport medium were processed immediately after arrival in our lab within a mean time of 22 hours. Given a transport time of 24-48 hours we evaluated the stability of the viral RNA over a 3 days period at room temperature and found a high stability of the results. This is mentioned in the section Material and Methods (lines 113-115 and 125-130) and Results (line 160). 

Point 2) “Please were samples tested in pools or singly? Why did authors not include oropharyngeal samples?“

All samples were tested individually, no pool-testing was performed. This is now mentioned in line 125. In Germany, oro- or nasopharyngeal sampling is recommended and we actually received both types of swabs. We subsumed everything under nasopharyngeal in the first version of our manuscript but now clarified the point in the revised manuscript (line 114).

Point 3) “Can't transporting samples at room temperature alter the quality of the genetic material and make the results uncertain in favour of the samples that would have survived the temperature better?“

This question could actually represent a bias and is a problem for all studies evaluating Ct values. In our opinion, one way to at least partially compensate for such preanalytical errors in single samples is to analyse large patient groups. Since, however, a large sample size confounds the significance levels, effect sizes should also be calculated as a consequence. Hence, if you consider pre-analytical errors and want to compensate for this with larger patient cohorts, this is another argument for effect sizes. We took up the reviewer's concerns and included a brief paragraph on this possible bias in the Discussion section (lines 218-230).

Point 4) “Indicate manufacturer and country for cobas® 8800 instrument“

Both details have been added (lines 122-123).

Point 5) “Write ORF1ab and E gene in italics“

This has been implemented (lines 124 and 125).

Point 6) “It is not only the age factor that has more than 2 groups, the setting factor also.“

The clinical setting was divided in out- and inpatients, which are two groups.

Point 7) “Please clarify if the comparison of patient percentages between Wild, Alpha and Beta types SARS-CoV-2 is for hospitalized patients only? Does this include only in-patients or also out-patients?“

The distribution of patients by wild type virus, Alpha and Beta VOC applies to the entire cohort. We added a sentence to the result section with the exact distribution of in- and outpatients for these three virus types (lines 164-166), so that this point should now be clarified.

Point 8) “Why still talk about Ct range between 15-36 when it had been declared that only samples with Ct value <30 are included in the study (line 102). Please clarify“

We further clarified our misleading formulation in the text (lines 111-112). 

Point 9) “The addition of all the cases (1013 wild type, 845 alpha VOC, 15 beta VOC) gives 1901 positive samples and not 1873 as declared in line 134. Please clarify“

We recalculated all data, but did not detect any mistakes. Adding 1013+845+15 gives 1873, but maybe we expressed something misleadingly. If so, we would be grateful for a hint. In addition, we discussed the results again in the authorship and came to the conclusion that the specification of eta squared is better suited than partial eta squared. Both work well as effect sizes in an ANOVA, but eta squared is considered easier to interpret. In order to show that both values differ only slightly and have no influence on the results or conclusions, we have included the table of the SPSS output as a supporting information (S1 Table) at the end of the manuscript.

In addition to the points mentioned above, we revised the manuscript according to the author's instructions and updated the reference list. We look forward to the review of our revised manuscript and hope that it is now considered acceptable for publication in PLoS One.

With kind regards,

Bernhard Zöllner

---

## [Editor Report · Decision Letter 1]

8 Jul 2022

The importance of effect sizes when comparing cycle threshold values of SARS-CoV-2 variants

PONE-D-22-07729R1

Dear Dr. Zöllner,

We’re pleased to inform you that your manuscript has been judged scientifically suitable for publication and will be formally accepted for publication once it meets all outstanding technical requirements.

Kind regards,

Etsuro Ito

Academic Editor

PLOS ONE

---

## [Editor Report · Acceptance letter]

12 Jul 2022

PONE-D-22-07729R1 

The importance of effect sizes when comparing cycle threshold values of SARS-CoV-2 variants 

Dear Dr. Zöllner:

I'm pleased to inform you that your manuscript has been deemed suitable for publication in PLOS ONE. Congratulations! Your manuscript is now with our production department. 

Kind regards, 

on behalf of

Prof. Etsuro Ito 

Academic Editor

PLOS ONE